# Evaluation of the Effectiveness of Children’s Dental Care Programs: A Retrospective Study

**DOI:** 10.3390/healthcare12070721

**Published:** 2024-03-26

**Authors:** Soo-Auk Park, Ji-Na Lim, Jae-Young Lee

**Affiliations:** 1Department of Dental Hygiene, College of Health Science, Dankook University, Cheonan 31116, Republic of Korea; sooauk5555@hanmail.net; 2Department of Public Health Science, Graduate School, Dankook University, Cheonan 31116, Republic of Korea; limyebang@dankook.ac.kr

**Keywords:** DMFT, child, adolescent dental care, cohort study

## Abstract

This study aimed to evaluate the long-term impact of children’s dental care programs on children and adolescents to reduce oral health inequalities. It measured and assessed the improvement effects of children’s dental care programs on the oral health of children and adolescents as part of the efforts to decrease oral health disparities in this age group. It included 406 individuals who participated in student and children’s dental care program between 2013 and 2019 at screening facilities in Gwangjin-gu, Seoul. A frequency analysis was conducted for demographic characteristics, and a binary logistic regression analysis was performed to identify factors influencing the prevalence of dental caries as the dependent variable. The data were analyzed using PASW Statistics with the statistical significance level set at α = 0.05. Regarding oral health status based on the frequency of participation in children’s dental care program for children and adolescents, participants with seven or more sessions had lower prevalence rates of dental caries, malocclusion, and periodontal disease than those with only one session. Second, when comparing oral health status in children’s dental care program between primary and adolescent age groups, individuals under continuous oral health care showed a decrease in permanent teeth affected by dental caries, dental caries prevalence, and malocclusion prevalence (excluding primary school age). Third, a binary logistic regression analysis revealed significant influences (*p* < 0.05) of the developmental stage and frequency of program participation on dental caries prevalence. Children’s dental care programs are essential for alleviating oral health inequalities among children and adolescents and preventing oral diseases. Furthermore, the developmental stage of children and the frequency of program participation are crucial factors in preventing oral conditions, such as dental caries.

## 1. Introduction

According to the National Health Insurance Service, dental caries has been identified as a prevalent oral disease in the past three years, making it a frequently diagnosed condition in dental care [1]. According to the ‘2022 Student Health Examination Analysis Data’ published by the Ministry of Education, the most common health issues among elementary, middle, and high school students in South Korea are visual impairment (55%) and dental caries (19%) [2]. The ‘Child Oral Health Status Survey’, conducted by the Ministry of Health and Welfare, indicates a gradual decrease in the number of permanent teeth affected by dental caries among 12-year-old children [3]. However, the World Health Organization’s (WHO) global average for 12-year-old permanent teeth affected by dental caries is 1.8, a level similar to that in Korea [4]. This emphasizes the need for continuous efforts to improve children’s oral health.

Childhood marks a crucial period in oral health management as it involves the transition from primary dentition to permanent dentition, known as the mixed dentition phase. Children’s oral health is closely linked to their nutritional intake, overall growth, and development. Habits formed during this period can potentially influence oral health issues in adulthood [5]. Sheiham [6] suggests that severe dental caries in children, leading to weight loss, could be attributed to dietary intake issues caused by tooth pain and growth inhibition due to chronic inflammation. Treating dental caries in preschool children not only boosts their growth and development, improving their quality of life, but also provides long-term cost savings on dental treatments through preventive oral care [7].

While South Korea’s health insurance system has rapidly developed to establish a comprehensive dental care guarantee for the entire population, its coverage remains relatively low compared to advanced nations [8]. The average household healthcare expenditure level is 36.8%, making South Korea the second-highest among OECD member countries [8]. Due to this relatively low coverage, healthcare inequality and medical poverty are on the rise, with 70% of the population being enrolled in private supplementary insurance [8]. Flores et al. [9] analyzed disparities in dental care utilization among U.S. children under 17, indicating that while some prevention of dental caries is achievable, it still disproportionately affects a small number of low-income children. Kim et al. [10] also argue that dental care inequality is evident, particularly in vulnerable populations, where difficulties in accessing dental care persist. A systematic literature review was conducted in Iran from 1994 to 2017 to assess the relationship between socioeconomic status and childhood dental caries. The study reported a significant inverse correlation between socioeconomic status and dental caries [11]. In Denmark, although the incidence of dental caries among 15-year-old children decreased from 1995 to 2013, relative inequalities increased across all socioeconomic categories [12]. In the Netherlands, despite the expansion of dental care coverage, economically vulnerable children were 1.5 times more likely to experience dental caries, and this inequality persisted into adolescence [13]. A systematic literature review spanning 1964 to 2018 examined dental caries among indigenous South American populations, including Brazil, Chile, Uruguay, and Venezuela. The results found that the mean dmft for 5-year-old children was 5.73 and the mean DMFT for 12-year-old adolescents was 3.14 [14]. This shows that dental caries is still an important public health problem. However, there are limited cases in advanced countries that provide public oral healthcare services, specifically for children and adolescents.

The children’s dental care program for children and adolescents was initiated to establish a primary dental care system that provides continuous preventive services with proven effectiveness in promoting oral health [15]. The program allows children and adolescents under 18 to register annually with a designated dentist, providing them with free treatment, a reimbursement system, and establishing a dental healthcare delivery framework to ensure ongoing management of oral health [16]. In South Korea, dental care predominantly focuses on treatment rather than prevention-centered oral health promotion [15,17]. However, the children’s dental care program adopts a preventive and education-centered approach that prioritizes sustainability by providing ongoing oral care. The inception of the children’s dental care program dates back to 2007, when the Korean Association of Oral Health Policy Research discussed and introduced the project [15], and by 2019, 43.8% of 1746 dental clinics and hospitals in Gyeonggi Province participated in the programs [18]. Despite being acknowledged as a vital welfare initiative, consistent challenges in budget allocation and a lack of funding increase resulted in difficulties highlighted by dental societies in Seoul, Gyeonggi Province, and Incheon City in October 2020 [19]. A review study on the children’s dental care program suggested that since the program is currently operated privately, there is a need for support in terms of funding, workers, and compensation for its operation, evaluation, and tracking surveys [16]. Ryu [20] argued that the dental care program for students and low-income children should be expanded by measuring its effectiveness in promoting oral health. However, the content determined by the regional consortia may vary, necessitating support measures and monitoring for consistency.

In contrast, the United Kingdom employs a registration system for dental care under the National Health Service (NHS), in which patients enroll in NHS-affiliated dental clinics to receive dental services. In this system, patients register with a knowledgeable dentist who provides comprehensive dental care, ranging from preventive services such as scaling and polishing to restorative treatments based on the patient’s oral health status and medical history. The UK operates an overarching program that clinically addresses dental care necessary for oral health promotion [21]. In contrast, South Korea, particularly some local governments, selects children and adolescents as eligible participants and enhances accessibility to dental care by providing prevention-centered dental services. The country operates a dedicated service that continually pursues oral health management to reduce inequality.

Oral diseases tend to initiate and intensify during childhood and adolescence and progressively worsen over time. Therefore, national strategies that address these age groups are required. Dental associations and civic organizations advocate implementing a primary care system for children and adolescents as an expansion of public oral health services. Informed by feedback on healthcare access challenges and the need for equitable service access, the primary care system’s goal is to enhance healthcare service quality, ensure more reliable health insurance, and reduce medical costs, thereby benefiting practitioners, residents, and the government by improving infrastructure, expanding coverage, and streamlining health insurance processes [22]. In particular, the children’s dental care program aligns with the direction of primary dental care, prioritizing preventive health promotion and making it necessary during the most effective period for preventive care: childhood and adolescence. Previous studies on children’s dental care program have mainly focused on the conceptualization, status, issues, and development plans for introducing the system. There is a significant gap in the research evaluating the long-term effects of children’s dental care program for improving children’s oral health through extended and systematic implementation.

Therefore, this study aimed to implement a children’s dental care program for low-income children and adolescents with limited long-term access to dental care services. Additionally, to ensure the future success of the program, this study aimed to establish a foundation for nationwide oral health initiatives by accumulating a database and building an oral health infrastructure. Furthermore, the goals are to increase the practice rate of oral health behaviors among children and adolescents, reduce economic costs, alleviate oral health inequalities, decrease the prevalence of dental caries, and provide continuous oral health management services to enhance and maintain the country’s oral health status globally.

## 2. Materials and Methods

### 2.1. Subjects

The study targeted the medical records of 406 individuals who participated in a children’s dental care program between January 2013 and December 2019 at public health centers (local health offices) in Gwangjin-gu, Seoul.

The region selection was based on the well-preserved state of the data in the database and the ability to obtain anonymized data through a data protection officer. The chosen institution was a local public entity, the health center, where data were acquired in collaboration with a personal information manager. A total of 406 individuals were included in the final analysis group for whom data analysis was feasible. This study is a pilot study and is representative, including a total of 4112 children in Gwangjin-gu, Seoul, with a final cumulative total of 406 patients over 7 years.

### 2.2. Data Collection and Procedures

Prior to conducting the research, approval was obtained from the Seoul National University Institutional Review Board (IRB No. S-D20200002). Data were collected in collaboration with the personnel responsible for the health center’s children’s dental care program. Data entered and reported by trained dentists between January 2013 and December 2019 were anonymized to ensure non-identifiability.

#### 2.2.1. Oral Examination and Health History

Health history obtained through the examination included information on oral health conditions (such as fractured teeth, tooth pain, gum pain, soft tissue pain, and bad breath), recent visits to medical clinics within the past year, toothbrushing habits, snacking, use of fluoride toothpaste, and oral health education (smoking and drinking habits).

#### 2.2.2. Oral Examination Report

Records were obtained based on oral examinations conducted by dentists with adequate education. Key components include the presence of restored teeth, dental caries, missing teeth, intraoral and soft tissue disorders, malocclusion, oral hygiene status, periodontal disease, Loe and Silness gingival index, temporomandibular joint disorders, enamel hypoplasia, and abnormalities in the wisdom teeth. Education was conducted based on the method used by WHO to educate and train investigators (research personnel) on oral examination methods.

#### 2.2.3. Dental Treatment

Oral health records and oral health activity questionnaires were collected through face-to-face surveys, and oral examinations were conducted by trained dentist. The key items included an oral hygiene examination (PHP examination), provision of oral health education, professional oral hygiene management, fluoride application, dental home care instructions, tartar removal, restorative treatment, root canal treatment, and extraction records.

#### 2.2.4. Yearly Oral Health Management Service Provision

Children’s dental care program conducted from 2013 to 2019 included oral health education, preventive care, and oral treatment. Oral health education, preventive care, and oral treatment were ranked in descending order. Professional oral hygiene management was the most common preventive care, followed by fluoride application, dental home care instructions, and tartar removal. Restorative treatment was the most frequently administered oral treatment, followed by root canal treatment, extraction, and other treatments. Also, the program consisted of additional contents such as oral health care methods, tooth brushing methods, use of oral hygiene products, and understanding of oral diseases, and the program was targeted at vulnerable groups in the region with low economic status.

### 2.3. Data Analysis

A frequency analysis was conducted to examine the demographic characteristics of the study participants. To identify the factors influencing the dependent variable, dental caries prevalence, a binary logistic regression analysis was performed using the enter method. The collected data were analyzed using PASW Statistics, version 23.0 (IBM Co., Armonk, NY, USA), and a statistical significance level of α = 0.05 was set.

In addition, to check the oral health status, the prevalence of dental caries, malocclusion, and periodontal disease were calculated. Prevalence was determined by comparing the prevalence (%) at the time of the survey by year of screening, based on the proportion of people who had the disease at that time. The prevalence of dental caries was calculated by dividing the sum of the number of people (exclude duplicates) with caries, fillings, and missing teeth by the total subjects and converting it into a percentage. The prevalence of malocclusion was calculated by dividing the number of people with malocclusion by the total subjects and converting it into a percentage. For the prevalence of periodontal disease, the number of people with periodontal disease was also divided by the total number of subjects and converted into a percentage. For the DMFT index, we divided the number of people with DMF into children and adolescents. For example, in 2019, when we participated once, we divided the number of people with DMF (327 people) into the cumulative number of people (406 people) and multiplied them by 100. In 2013, when we participated seven times, the number of people with DMF (30) was divided into 49 people, and the results were calculated by multiplying by 100.

## 3. Results

### 3.1. Demographic Characteristics of Subjects

A total of 406 study participants participated in the children’s dental care program between 2013 and 2019. The demographic characteristics of the study participants included 209 boys (51.5%), 197 girls (48.5%), 292 school-aged individuals (71.9%), and 114 adolescents (28.1 people) (Table 1).

The interviews revealed that among those with oral health problems, 68 (16.7%) had fractured teeth, 125 (30.8%) had tooth pain, 91 (22.4%) had gum pain, 38 (9.4%) had soft tissue pain in the tongue and cheeks, and 115 (28.3%) patients had symptoms of bad breath. Regarding whether they had visited hospitals or clinics in the past year, 286 (70.4%) had visited. The most frequent number of times study subjects brushed their teeth per day was 2 (157 people (38.7%)), 232 people (57.1%) frequently consumed snacks such as carbonated drinks, and 64 people (15.8%) used fluoride toothpaste containing fluoride. They responded that they used toothpaste approved for use. Regarding oral health education, 133 participants (32.9%) said they had received education about smoking, and 192 (47.3%) said they had received education about drinking.

An examination revealed that 142 participants (35.0%) had decayed teeth, 302 (74.4%) had filled teeth, and 7 (1.7%) had missing teeth. None of the patients had stomatitis or soft tissue disease (0.0%); 61 patients (15.0%) had malocclusion; 288 (70.9%) had average oral hygiene; and 85 (20.9%) had excellent oral hygiene. There were 32 people (7.9%) who had periodontal disease; none had temporomandibular joint abnormalities (0%); 24 (5.9%) had tooth attrition, and none had wisdom tooth abnormalities (0.0%).

### 3.2. Oral Health Status by Number of Participations in Children’s Dental Care Program

We presented the cumulative number of participants and oral health status of the beneficiaries of children’s dental care programs by the number of times they participated (Table 2). The cumulative number of participants was 406 participants for the 1st time, 287 participants for the 2nd time, 224 participants for the 3rd time, 176 participants for the 4th time, 128 participants for the 5th time, 75 participants for the 6th time, and 49 participants for the 7th time. As the number of project participants increased, there was a tendency for the prevalence of dental caries, malocclusion, and periodontal disease to decrease among beneficiaries who continued to be managed.

### 3.3. Effect of Improving Oral Health Indicators according to the Implementation of the Children’s Dental Care Program

The improvement effects resulting from implementing the children’s dental care program were divided into school age and adolescence and presented by the oral health index (Table 3). The results were compared for first-time patients (state before implementation) and 7th cumulative participants. A total of 292 children and 114 adolescents were compared.

Before carrying out the program, the number of permanent teeth with caries experience at school age was 0.32; however, after continuous management, the effect decreased to 0.26, and in adolescence, it also decreased from 2.73 to 0.87. The prevalence of dental caries decreased from 36.1% to 29.9% in school-aged children and from 72.8% to 38.8% in adolescents. The prevalence of malocclusion increased from 12.0% to 16.8% in school-aged children and decreased from 27.2% to 13.2% in adolescents.

### 3.4. Factors Affecting the Prevalence of Dental Caries

A binomial logistic regression analysis was performed to identify the factors that had a statistically significant impact on the prevalence of dental caries. The independent variables were continuous variables such as age, number of participations, and number of daily toothbrushing sessions, and categorical variables such as gender, developmental stage, snack intake, use of fluoride toothpaste, and oral health education; the dependent variable was the prevalence of dental caries (Table 4).

The analysis showed a statistically significant effect on the development stage and number of program participations (*p* < 0.05). In the developmental stage, adolescents were more likely to have dental caries than school-aged adolescents (*p* < 0.001, B = −1.371). Since B is negative and OR is less than 1, the probability of having caries in school age was 0.254 times higher than in adolescence. In other words, the likelihood of having dental caries in school-aged children was 74.6% lower than that in adolescents (OR = 0.254). Participation in the program showed that the higher the number of participations, the greater the likelihood of no dental caries (*p* < 0.05, B = −0.182). Since B was negative and OR was less than 1, when the number of participations increased by 1, the odds of the internal value of 1 of the dependent variable increased by 0.834 times. Thus, when the number of participations increased by 1, the likelihood of having dental caries decreased by 16.6% (OR = 0.834).

## 4. Discussion

Korea’s total fertility rate is 0.78 based on 2022 National Statistical Office data, making it the lowest rate among OECD member countries [23]. Consequently, medical infrastructure, such as medical personnel and medical institutions dedicated to children and adolescents, is weakening, and the need for a children’s healthcare system that considers the activation of dedicated treatment for children and adolescents and the establishment of a medical delivery system continues to increase. Following this trend, the government recently declared a national responsibility system for children and presented national tasks to strengthen public health care for children [24]. Introducing children’s dental doctors was considered a way to strengthen the comprehensive management of childhood health risk factors, and the National Health Insurance Policy Deliberation Committee discussed the children’s dental doctor pilot program as an agenda item and decided on specific details. The Ministry of Health and Welfare calculates the fee for the pilot program for children and adolescent dentists in six-month increments [25]. However, over the past four years, the burden of treatment costs on the public has been increasing as diseases related to the oral cavity have ranked high [1]. Accordingly, the need for oral healthcare has been emphasized to reduce the economic burden.

To overcome the limitations of previous research on primary care dentists for children and adolescents, this study implemented a mid- to long-term dental care program, targeting children and adolescents from low-income families with limited access to dentistry. This study was designed to evaluate the effectiveness of policies for continuous children’s oral health care. It is a representative sample of children living in a single region who have participated for many years. Most of the research subjects who participated in this study’s children and adolescent dental care program were male (51.5%) and of school age (71.9%) (Table 1).

First, looking at the main results, the prevalence of dental caries, malocclusion, and periodontal disease were all found to decrease according to the number of participation times in the children and adolescent dental care program among participants who participated seven times compared with those who participated once (Table 2). This was similar to the results of a previous study that analyzed the prevalence of dental caries by life cycle, which showed that the timing of dental visits and lack of dental treatment significantly impacted the prevalence of dental caries in children and adolescents [26]. In the results of previous studies, if the number of visits to the dentist or lack of treatment is considered in the number of program participants, this can be interpreted to mean that the prevalence of dental caries decreases when frequent visits to the dentist and dental treatment are performed. In addition, according to many studies, it has been reported that multiple customized treatments based on individual caries risk, such as this children’s dental care program, may be more beneficial to oral health than a one-time approach [27]. In a study involving 483 adolescents, Gabris et al. [28] reported a significantly higher rate of caries experience in cases with malocclusion compared to those without. Stahl and Grabowski [29] conducted a study on 7639 children aged 7 to 10 during the mixed dentition period, suggesting that while generalizing the association between malocclusion and dental caries prevalence is challenging, specific forms of malocclusion are related to caries prevalence. Based on these results, there appears to be a correlation between malocclusion and the prevalence of dental caries. In the present study, as the frequency of participation in the programs increased, the prevalence of dental caries decreased. This could be interpreted as a decrease in the prevalence of malocclusion, which is correlated with the prevalence of dental caries. Yun and Suh [30] argued that the number of dental outpatient visits for patients with periodontal disease increased after implementing the scaling reimbursement policy, suggesting that as individuals visit the dentist more frequently, the prevalence of periodontal disease decreases due to increased periodontal treatment. Similarly, in this study, a higher frequency of dental visits was associated with a lower prevalence of periodontal disease.

The following text compares the oral health-related indicators of the children and adolescent dental care program by dividing them into school-age and adolescent groups. The results showed that the number of permanent teeth with caries, dental caries prevalence, and malocclusion prevalence (excluding school age) showed continued management compared with those receiving the first treatment. It decreased in one subject (Table 3). However, the prevalence of malocclusion was found to increase in school-aged subjects who received continuous care compared to those who received first-time treatment (Table 3). In school-aged children, the probability of malocclusion occurs between the ages of 6 and 12 years, when the primary teeth fall out and the permanent teeth begin to erupt. This may have influenced the results of the present study because it is a mixed dentition period with increasing height. Furthermore, since oral diseases are irreversible and accumulate over time, future research will be needed to determine whether the treatments and interventions of children’s dental care have direct consequences.

Finally, as a result of the binomial logistic regression analysis conducted to confirm the effect on the prevalence of dental caries, we found that the developmental stage and number of program participations had a significant effect (*p* < 0.05) (Table 4). In the developmental stage, adolescents were 74.6% more likely to have dental caries than school-age adolescents (*p* < 0.001), and as the number of program participations increased by one, the likelihood of not having dental caries increased by 16.6% (*p* < 0.05). This is because permanent teeth do not regenerate, and dental caries is an accumulating disease; therefore, the experience of dental caries tends to increase with age [31]. It is interpreted that adolescents have relatively more dental caries than school-age children. In addition, according to previous studies, low average monthly household income, high rates of unmet dental care, and high oral health inequality are associated with an increase in the prevalence of dental caries [31,32]. Therefore, if participation in a child dental care program increases, the prevalence of dental caries also increases. This is interpreted as a decrease. Accordingly, public intervention is needed to increase access to dentistry for low-income individuals, which is expected to contribute to resolving oral health inequalities.

The children and adolescent dental care program started in 2019 when COVID-19 first occurred, was halted in 2020 due to COVID-19, and resumed in 2022 for 4th and 5th grade elementary school students. As of March 2023, the utilization rate of the child and adolescent dental doctor pilot program promoted in Gwangju Metropolitan City and Sejong City was 4924 (25.1%) out of 19,589 target children. As of 2021, Seoul City (70.8%), compared to Gyeonggi-do (87.2%), was only one-third. To resolve this low participation rate, active measures are needed to expand services, such as reducing out-of-pocket costs, expanding the target area and age, and providing follow-up treatment for those with symptoms.

As this study tracked the number of participants from 2019, the limitation was that it could not confirm whether the participants participated continuously. However, as the number of participants increased, the caries experience permanent tooth index (DMFT index) decreased. The children and adolescent dental care program provides both prevention and treatment programs. Prevention programs were generally implemented for all participants; however, treatment programs were conducted only for those in need. Differences may have occurred depending on the effectiveness of these treatments, which may have affected the results.

Nevertheless, the children and adolescent dental care program focuses on prevention and is a program to alleviate oral health inequality and determine whether accessibility to low-income families has increased in the mid- to long-term or whether it continues to improve the oral health status and oral health of children in the participating and comparison groups. To analyze changes in behavior, such as level (knowledge, perception, attitude, and behavior), it is necessary to track, observe, and evaluate the effect in the mid- to long-term. Accordingly, this study was conducted over a period of 7 years from 2013 to 2019. It is significant that meaningful results were derived from the program performance data.

The children and adolescent dental care program is a program that must be established and implemented in elementary schools across the country, and a dental doctor who can manage both the lower and upper grades is required. Accordingly, by setting short- and mid- to long-term goals, services should be provided starting with children in lower grades, and future-oriented and active oral care should be implemented by reaching older children, teenagers, and adults as they grow. Additionally, based on the caries risk and questionnaire results, the current oral conditions of children and adolescents must be carefully investigated to establish a protocol appropriate for the current situation. The program’s effectiveness must be evaluated through the provision of appropriate oral health services and comparative analysis. In addition, we are strengthening the consistency and accountability of the policy implementation process for stakeholders, such as local governments, the Ministry of Health and Welfare, and dentists who participate in and operate the children and adolescent dental care program, and establishing cooperative governance among stakeholders. Therefore, this effect should be evaluated. Additionally, subsequent research will be needed to reduce the incidence of new caries through prevention programs rather than overall caries experience.

## 5. Conclusions

This study aimed to measure and evaluate the improvement effect of a dental care program on the oral health of children and adolescents by implementing a child and adolescent dental care program over the mid- to long-term to reduce oral inequality among children and adolescents. The main conclusions are as follows:It was found that the group that continued to participate in this program for many years had a lower prevalence of dental caries, malocclusion, and periodontal disease compared to the group that started participating in the program for the first time.As a result of comparing the oral health-related indicators of the children’s dental care program by dividing them into school age and adolescence, it was found that all of them decreased in those who received continuous care compared to those who had their first visit. This decreased in one subject. In contrast, the prevalence of malocclusion increased in school-age subjects who received continuous oral health care for many years compared to those who received the first treatment.Binomial logistic regression analysis to determine the effect on dental caries prevalence revealed that the developmental stage and number of program participations had a significant effect (*p* < 0.05).

These results confirm that a dental care program for children and adolescents is necessary to alleviate oral health inequalities and prevent oral diseases. Future research will require an in-depth evaluation of the effectiveness of dental-care programs for children and adolescents.

## Figures and Tables

**Table 1 healthcare-12-00721-t001:** Demographic characteristics of study subjects (N = 406).

Characteristic	Division	N	%
Personal characteristics			
Sex	Boys	209	51.5
	Girls	197	48.5
Age	Child (6–12 years old)	292	71.9
	Adolescence (13–18 years old)	114	28.1
Questionnaire results			
Oral health problems	Fractured tooth	68	16.7
	Tooth pain (hyperesthesia)	64	15.8
	Tooth pain (throbbing, throbbing)	61	15.0
	Gum pain	91	22.4
	Soft tissue pain (throbbing tongue and cheek)	38	9.4
	Halitosis	115	28.3
Have you visited a hospital or clinic in the past year?	Yes	286	70.4
	No	120	29.6
Number of toothbrushing times per day	0	11	2.7
	1	68	16.7
	2	157	38.7
	3	123	30.3
	4	36	8.9
	5	6	1.5
	6	5	1.2
Snacking	Yes	232	57.1
	No	174	42.9
Use of fluoride toothpaste	Yes	64	15.8
	No	341	84.0
Oral health education (smoking)	Yes	133	32.9
	No	273	67.2
Oral health education (drinking)	Yes	192	47.3
	No	214	52.7
Examination results			
People with filled teeth	Yes	302	74.4
	No	104	25.6
People with decayed teeth	Yes	142	35.0
	No	264	65.0
People with missing teeth	Yes	7	1.7
	No	399	98.3
Stomatitis and soft tissue diseases	Yes	0	0.0
	No	406	100.0
Malocclusion	Yes	61	15.0
	No	345	85.0
Oral hygiene status	Excellent	85	20.9
	Average	288	70.9
	Needs Improvement	33	8.1
Periodontal disease	Yes	32	7.9
	No	374	92.1
Temporomandibular joint abnormalities	Yes	0	0.0
	No	406	100.0
Tooth attrition and abrasion	Yes	24	5.9
	No	382	94.1
Wisdom teeth abnormalities	Yes	0	0.0
	No	406	100.0

**Table 2 healthcare-12-00721-t002:** Oral health status by number of participations in children’s dental care program.

Participations		People with DMFT (n)				Prevalence (%)			Participants (n)
Year	N	D	M	F	DMF *	Caries	Malocclusion	Periodontitis	Cumulative
2013	7th	12	0	41	30	61.2	2.0	2.0	49 (12.1)
2014	6th	17	1	51	41	54.7	5.3	2.7	75 (18.5)
2015	5th	49	3	75	68	53.1	8.6	6.3	128 (31.5)
2016	4th	62	4	100	91	51.7	8.5	9.2	176 (43.3)
2017	3rd	43	2	131	109	48.7	6.3	3.1	224 (55.2)
2018	2nd	62	9	176	175	61.0	13.6	5.2	287 (70.7)
2019	1st	142	7	302	327	80.5	15.0	7.9	406 (100.0)
Cumulative		245	26	876	841				1345

* Excluding duplicate people.

**Table 3 healthcare-12-00721-t003:** Effect of improving oral health-related indicators according to the implementation of the children’s dental care program.

Oral Health-Related Indicators	Status before Performance (First Visit)		Effects after Implementation (Continued Management)
DMFT index (n)	Child (6–12 years old)	0.32	0.26
	Adolescence (13–18 years old)	2.73	0.87
Dental caries prevalence (%)	Child (6–12 years old)	36.1	29.9
	Adolescence (13–18 years old)	72.8	38.8
Malocclusion prevalence (%)	Child (6–12 years old)	12.0	16.8
	Adolescence (13–18 years old)	27.2	13.2

**Table 4 healthcare-12-00721-t004:** Impact on dental caries prevalence.

Factor	B	SE	Wald	*p*	OR	95% CI	
						LLCI	ULCI
^+^ Sex *Male	0.121	0.219	0.303	0.582	1.128	0.734	1.734
^+^ Developmental stage *Child	−1.371	0.416	10.868	0.001 **	0.254	0.112	0.574
Age	−0.128	0.075	2.916	0.088	0.880	0.760	1.019
Number of participations	−0.182	0.076	5.667	0.017 *	0.834	0.718	0.968
Number of toothbrushing times per day	−0.095	0.103	0.854	0.355	0.909	0.744	1.112
^+^ Snacking *Yes	−0.282	0.224	1.587	0.208	0.755	0.487	1.170
^+^ Use of fluoride toothpaste *Yes	0.311	0.317	0.968	0.325	1.365	0.734	2.539
^+^ Oral health education (smoking) *Yes	−0.453	0.237	3.670	0.055	0.635	0.400	1.011
^+^ Oral health education (drinking) *Yes	−0.056	0.220	0.066	0.798	0.945	0.614	1.454

Logistic regression analysis * *p* < 0.05, ** *p* < 0.01, ^+^ reference group: Sex *Female, Developmental stage *Adolescence, Snacking *No, Use of fluoride toothpaste *No, Oral health education (smoking, drinking) *No.

## Data Availability

The datasets used or analyzed during the current study are available from the corresponding author on reasonable request.

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
