# Peer review of "Evaluation of the Effectiveness of Children’s Dental Care Programs: A Retrospective Study"

_healthcare, 2024, doi:10.3390/healthcare12070721_

Round 1

Reviewer 1 Report

Comments and Suggestions for Authors

The study is original retrospective research on improvement effects of dental care program (prevention and treatment) for children and adolescents in Korea. The research is interesting and important as questions and problems presented are actual worldwide. Major remarks: it is necessary to explain methodology in detail. We do not know whether the sample is representative. Authors should describe what standards were used for evaluating oral hygiene and periodontal disease. It is necessary to improve citation and correct errors made in translation which diminish clearness and readability of the content.

Title

In the title “project” is mentioned, while in the manuscript “program” is used. Please choose one term and use it consistently throughout the whole manuscript.

Introduction

Inaccurate citation is present through whole Introduction & Discussion sections, which must be corrected.

Studies/countries/authors are named in the text, but references’ numbers given in brackets include more unrelated references. Line 57: “The average household healthcare expenditure level is 36.8%, making South Korea the second-highest among OECD member countries [8–9]. Due to this relatively low coverage, healthcare inequality and medical poverty are on the rise, with 70% of the population being enrolled in private supplementary insurance [8–10].” Reference No. 9 is about racial disparities in medical and oral health in US children, while text is on situation in South Korea, so only references No. 8 and 10 should be included.

“Flores et al. [9–11] analysed disparities in dental care service utilization

among U.S. children under 17, indicating that while some prevention of dental caries is achievable, it still disproportionately affects a small number of low-income children.” Here only reference No. 9 should be included.

Line 73: “A systematic literature review spanning 1964 to 2018 examined dental caries among indigenous South American populations, including Brazil, Chile, Uruguay, and Venezuela. The study estimated the DMFT index for individuals aged 5 to 74 years and reported the results [14–16].” Here authors are mentioning a research which is reference No. 14 (again inaccurate citation with inclusion of references No. 15 and 16). The authors are stating that “research on childhood dental caries is prevalent internationally” which is unimportant information but omit to describe the finding of the research. Please tell us what is the trend in dental caries in South American populations.

References 18 and 19 should contain title of the online article in English.

The last paragraph of the Introduction is a mixture of the goals of the preventive program and goals of the present research which is evaluation of the program’s effectiveness. Please describe goals of the preventive program separately, and reserve the last paragraph for the aim of the present research.

Materials and Methods

The research included 406 individuals. Please state what was the total number of individuals included in the program in that period, and discuss the representativeness of the sample. Did the program include children living in villages? Were the children with special needs included in the program? In line 354 “control groups” are mentioned. Have some children been left out of the program to be a control group? Please state these details.

Line 160: “Records were obtained based on oral examinations conducted by dentists with adequate education.” Please describe what is “adequate education”.

Line 166: “Records were orally administered to students and paediatric dental patients based on health history and oral examinations.” The content is not clear, probably this resulted from wrong translation. Please correct the sentence. The title 2.2.3. should be Dental Treatment.

Authors should describe what standards were used for evaluating oral hygiene and periodontal disease. World Health Organization does not consider probation of periodontal pockets in individuals under 15 yrs of age, so it would be interesting to know how did you evaluate periodontal disease.

Results

Consider replacing “men” and “women” with “boys” and “girls”.

Table 1. needs change of row width, as presently number 209 (males) is presented in two rows as 20 and 9.

Table 2. row categories are not clear. Please describe in detail. Lines 213-219 are not clear, and there are errors probably due to wrong translation.

Please describe in detail how calculation of DMFT decrease was made, so that other investigators can follow the same procedure.

Line 228 “the oral health index (Table 3)” is mentioned, while in Table 3 there is a column titled "Oral health-related indicators”.

Lines 238, 247 & 248, please replace “participants” with “participations”.

Discussion

Several terms are used in the manuscript: “children and adolescent dentists” and “primary care dentists for children and adolescents”. Please clarify is it a specialist of paediatric dentistry, or a general dentist or family dentist who is engaged in the program? Also, is paediatric dentistry a recognized speciality in Korea?

Lines 270-277, multiple language and content errors due to bad translation. Example: “oral health business”.

Lines 278-281 should be cut out as this is repetition of Material and Methods information. Instead, authors should discuss representativeness of the sample.

Please clarify what is “continued management” (lines 311 & 386).

Lines 383-387, the sentence needs to be improved as it is not clear.

Comments on the Quality of English Language

There are many errors made in translation which diminish clearness and readability of the content.

Lines 270-277, multiple language and content errors due to bad translation. Example: “oral health business”.

Author Response

REVIERW 1

The study is original retrospective research on improvement effects of dental care program (prevention and treatment) for children and adolescents in Korea. The research is interesting and important as questions and problems presented are actual worldwide. Major remarks: it is necessary to explain methodology in detail. We do not know whether the sample is representative. Authors should describe what standards were used for evaluating oral hygiene and periodontal disease. It is necessary to improve citation and correct errors made in translation which diminish clearness and readability of the content.

Answer)

Thank you so much for your quick and good feedback. I have modified the menu script to reflect all of the reviewer's opinions.

1)

Title

In the title “project” is mentioned, while in the manuscript “program” is used. Please choose one term and use it consistently throughout the whole manuscript.

Answer)

Reflecting the reviewer's opinion, We unified not only the title but also the manuscript as “program”.

2)

Introduction

Inaccurate citation is present through whole Introduction & Discussion sections, which must be corrected. Studies/countries/authors are named in the text, but references’ numbers given in brackets include more unrelated references.

Line 57: “The average household healthcare expenditure level is 36.8%, making South Korea the second-highest among OECD member countries [8–9]. Due to this relatively low coverage, healthcare inequality and medical poverty are on the rise, with 70% of the population being enrolled in private supplementary insurance [8–10].” Reference No. 9 is about racial disparities in medical and oral health in US children, while text is on situation in South Korea, so only references No. 8 and 10 should be included.

Answer)

Reflecting the reviewer's opinion, reference 9 was deleted and only references 8 and 10 were included. (Ln57)

3)

“Flores et al. [9–11] analysed disparities in dental care service utilization

among U.S. children under 17, indicating that while some prevention of dental caries is achievable, it still disproportionately affects a small number of low-income children.” Here only reference No. 9 should be included.

Answer) Reflecting the reviewer's opinion, only reference 9 was included, and references 10 and 11 were deleted. (Ln61)

4)

Line 73: “A systematic literature review spanning 1964 to 2018 examined dental caries among indigenous South American populations, including Brazil, Chile, Uruguay, and Venezuela. The study estimated the DMFT index for individuals aged 5 to 74 years and reported the results [14–16].” Here authors are mentioning a research which is reference No. 14 (again inaccurate citation with inclusion of references No. 15 and 16). The authors are stating that “research on childhood dental caries is prevalent internationally” which is unimportant information but omit to describe the finding of the research. Please tell us what is the trend in dental caries in South American populations.

Answer) References 15 and 16 were deleted, and information on dental caries trends in the South American population was added. (Ln74-78)

5)

References 18 and 19 should contain title of the online article in English.

Answer) It was replaced with a thesis based on the reviewer's opinion. (Ln457-460)

6)

The last paragraph of the Introduction is a mixture of the goals of the preventive program and goals of the present research which is evaluation of the program’s effectiveness. Please describe goals of the preventive program separately, and reserve the last paragraph for the aim of the present research.

Answer)

To reflect the reviewer's opinion, we have deleted the content evaluating the effectiveness of the program. (Ln129-131)

7)

Materials and Methods

The research included 406 individuals. Please state what was the total number of individuals included in the program in that period, and discuss the representativeness of the sample. Did the program include children living in villages? Were the children with special needs included in the program? In line 354 “control groups” are mentioned. Have some children been left out of the program to be a control group? Please state these details.

Answer) This study is a pilot study and is representative, including a total of 4,112 children in Gwangjin-gu, Seoul, with a final cumulative total of 406 patients over 7 years. (Ln147-149)

The control group was misspelled and was corrected to comparison group. (Ln362-363)

8)

Line 160: “Records were obtained based on oral examinations conducted by dentists with adequate education.” Please describe what is “adequate education”.

Answer)

“Education was conducted based on the method used by WHO to educate and train investigators (research personnel) on oral examination methods.” The content has been added. (Ln167-169)

9)

Line 166: “Records were orally administered to students and paediatric dental patients based on health history and oral examinations.” The content is not clear, probably this resulted from wrong translation. Please correct the sentence. The title 2.2.3. should be Dental Treatment.

Answer)

“Health records and questionnaires were collected through face-to-face surveys, and oral examinations were performed.” It was modified to. (Ln172-173)

Reflecting the reviewer's opinion, 2.2.3. The title has been changed to Dental Treatment. (Ln171)

10)

Authors should describe what standards were used for evaluating oral hygiene and periodontal disease. World Health Organization does not consider probation of periodontal pockets in individuals under 15 yrs of age, so it would be interesting to know how did you evaluate periodontal disease.

Answer)

This information has been added to the methods section. The study was conducted based on the Löe&Silness gingival index (GI). (Ln165-166)

11)

Results

Consider replacing “men” and “women” with “boys” and “girls”.

Answer)

It was revised to “boys” and “girls” to reflect the reviewer’s opinion. (Ln206 and Table 2)

12)

Table 1. needs change of row width, as presently number 209 (males) is presented in two rows as 20 and 9.

Answer) The row width has been changed to reflect your opinion. (Table 1)

13)

Table 2. row categories are not clear. Please describe in detail. Lines 213-219 are not clear, and there are errors probably due to wrong translation.

Answer) Reflecting the reviewer's opinion, categories that were not necessary in context were deleted, and lines 213-219 were moved to the Data Analysis section. (Table 2)

14)

Please describe in detail how calculation of DMFT decrease was made, so that other investigators can follow the same procedure.

Answer)

Training was conducted according to WHO standards, and the contents of this were reinforced in the material & methods. (Ln167-169)

15)

Line 228 “the oral health index (Table 3)” is mentioned, while in Table 3 there is a column titled "Oral health-related indicators”.

Answer) The terminology section was reviewed throughout the document, including oral health status. (Table 3)

16)

Lines 238, 247 & 248, please replace “participants” with “participations”.

Answer) It was revised to “participations” to reflect the reviewer’s opinion. (Ln252, 261-262)

17)

Discussion

Several terms are used in the manuscript: “children and adolescent dentists” and “primary care dentists for children and adolescents”. Please clarify is it a specialist of paediatric dentistry, or a general dentist or family dentist who is engaged in the program? Also, is paediatric dentistry a recognized speciality in Korea?

Answer) The terminology was unified as children's dental care service program, and the dentists participating in the program were operated by trained dentists, including pediatric dentists, although a dentist specialist system is in operation in Korea.

18)

Lines 270-277, multiple language and content errors due to bad translation. Example: “oral health business”.

Answer) In general, terminology has been reorganized.

19)

Lines 278-281 should be cut out as this is repetition of Material and Methods information. Instead, authors should discuss representativeness of the sample.

Answer) All children in Seoul were targeted in the Gwangjin-gu constituency, and since all children were targeted at vulnerable groups with low economic status, information on representativeness was added to the discussion and material & methods. (Ln287-289)

20)

Please clarify what is “continued management” (lines 311 & 386).

Answer) Details of the dentist's care program are described in Materials and Methods. (Ln165-167, 185-188)

21)

Lines 383-387, the sentence needs to be improved as it is not clear.

Answer) “As a result of comparing the oral health-related indicators of the children's dental care service program by dividing them into school age and adolescence, it was found that all of them decreased in those who received continuous care compared to those who received their first visit.” It was modified like this. (Ln393-396)

22)

Comments on the Quality of English Language

There are many errors made in translation which diminish clearness and readability of the content.

Answer) Additional proofreading was performed through Editage, a professional English translation.

23)

Lines 270-277, multiple language and content errors due to bad translation. Example: “oral health business”.

Answer) A general correction was made for the relevant vocabulary selection, and additional corrections were performed through professional English translation Editage.

Reviewer 2 Report

Comments and Suggestions for Authors

Methodology not thorough enough: How was the number of participations calculated? Each participation was the same? at each meeting the patients received the same treatment? As described in lines 166-170?  

The use of concepts is misleading: line 166 for example, lines 160-161: adequate education? Line 171: Oral Health Management Service Provision? who needed Management Services provision?

Table 2 not clear: All the 49 who had 7 participations are included in the 1 participation too? Else how are 406 participants with 1 participation?

Line 230 dividing which number by which number? 

How malocclusion improved by participation if no orthodontics were performed?

Author Response

REVIERW 2

ANSWER) We would like to thank the reviewer for his valuable comments on the submitted paper, and we have made changes as much as possible to reflect the comments.

1)

Methodology not thorough enough: How was the number of participations calculated? Each participation was the same? at each meeting the patients received the same treatment? As described in lines 166-170?

ANSWER) We have added information about the methodology to reflect the opinions of the reviewers. The number of participation in the program was based on an annual basis, and the participation of the child and research ethics were reported in advance to the healthcare editorial board and sent to the IRB for review.

2)

The use of concepts is misleading: line 166 for example, lines 160-161: adequate education? Line 171: Oral Health Management Service Provision? who needed Management Services provision?

ANSWER) I received a program treatment course and added the content to the materials and methods. Additional educational content: Contents such as oral health care methods, tooth brushing methods, use of oral hygiene products, and understanding of oral diseases were organized and added, and the program is included in the corresponding program. It was conducted for vulnerable groups in the region with low economic standards. (Ln167-169, 185-188)

3)

Table 2 not clear: All the 49 who had 7 participations are included in the 1 participation too? Else how are 406 participants with 1 participation?

ANSWER) The student who participated 7 times is a friend who has participated consistently for 7 years, from several years to several years, and all program courses were conducted and reinforced more than twice a year. (Table 2)

4)

Line 230 dividing which number by which number?

ANSWER) An oral examination was performed based on WHO standards, and the DMFT index was used to indicate the caries status of the teeth. (Table 3)

5)

How malocclusion improved by participation if no orthodontics were performed?

ANSWER) It appears that the data on the improvement rate of malocclusion after the correction you mentioned will need to be strengthened through follow-up studies.

Round 2

Reviewer 2 Report

Comments and Suggestions for Authors

The paper deals with "the effectiveness of children's dental care program" not service. 

Author Response

Comments on review opinions were written point to point and uploaded according to the designated format. thank you

REVIERW 2

The paper is fine now except the unnecessary "service" in the title and in the first half of the paper. The paper deals with "the effectiveness of children's dental care program" not service.

ANSWER) Thank you. In consideration of the reviewer's opinion, "service" was removed and everything was replaced with "dental care program."